# Surface Characteristics of Polymers with Different Absorbance after UV Picosecond Pulsed Laser Processing Using Various Repetition Rates

**DOI:** 10.3390/polym12092018

**Published:** 2020-09-04

**Authors:** Seung Sik Ham, Ho Lee

**Affiliations:** 1Institute for Nano Photonics Applications, Kyungpook National University, Daegu 41566, Korea; ssh13@knu.ac.kr; 2School of Mechanical Engineering, Kyungpook National University, Daegu 41566, Korea; 3Laser Application Center, Kyungpook National University, Daegu 41061, Korea

**Keywords:** polyimide, polyethylene terephthalate, picosecond laser, heat accumulation effect, carbonization phenomenon

## Abstract

We experimented with two polymer materials with different ultraviolet (UV) wavelength absorption characteristics, which are commonly used in flexible devices, by applying an ultrashort-pulsed laser of a 355-nm UV wavelength for 10 ps. The laser parameters studied were pulse repetition rate, laser irradiation method, and laser power condition. Previous studies using polyethylene terephthalate (PET), which does not exhibit linear absorption at a UV wavelength, have focused on processing trends resulting in minimal collateral damage around the laser-induced ablation. However, our results showed a trend of accumulating such damage irrespective of the laser parameters. Meanwhile, polyimide (PI) exhibited a completely different behavior depending on the laser parameters. At low pulse repetition rates, minimal collateral damage was observed, whereas at high repetition rates, the morphology varied considerably. The electrical characteristics of the laser-processed materials were found to be correlated with the variations in morphology. In the case of PI, such variations in electrical resistance and morphology indicated that the material was carbonized. The findings of this study are expected to provide a useful reference when selecting parameters for the laser processing of similar polymer materials.

## 1. Introduction

In recent years, research interest in lightweight flexible devices that can change shape freely has increased [1,2,3]. Polymers are widely used as substrates in these flexible devices owing to their unique properties. Among such polymer materials, polyimide (PI) and polyethylene terephthalate (PET) are often used because of their excellent chemical stability and heat resistance [4,5,6].

Polymer processing using lasers has been researched extensively, and the source used for the first time in the industrial field is the CO_2_ laser. CO_2_ lasers have been commonly used because their wavelength of 9.2–11.4 μm, in the mid-infrared region, which works as a continuous wave, has a high-power output, and can be absorbed even by non-metal and transparent materials. CO_2_ laser processing of polymer materials has been employed for manufacturing microfluidic devices [7,8,9,10], polymer bonding [11,12,13,14], and micro-processing of printed circuit boards (PCBs) and flexible printed circuit boards (F-PCBs) [15,16]. However, CO_2_ lasers have a large peripheral heat effect and frequently induce secondary damage because the interaction mechanism between the laser and the material is dominated by thermal reactions, and it is difficult to manage because gas is used as the laser medium. Another disadvantage is the difficulty in selecting and managing components for configuring the optical system for laser beam transmission because of the linear absorption in general optical components.

Other laser sources mostly use the UV wavelength, which causes linear absorption in polymers. As UV lasers have much larger photon energy than CO_2_ lasers, they induce less thermal damage and less secondary damage. In other words, they enable more precise processing than CO_2_ laser processing. UV wavelength lasers have been used in PCB and F-PCB processing [1,17,18,19], lithography [20,21,22,23], indium tin oxide (ITO) patterning [24,25,26], and 3D printing [27,28]. However, UV lasers have disadvantages in general cutting and welding applications, such as being time-consuming, because most of them are constrained to operate at low power. In addition, excimer lasers, which use a gas gain medium, are difficult to manage although the beam quality is good, while, by contrast, solid-state lasers exhibit poor quality compared to excimer lasers.

The development of ultrashort-pulsed lasers has drawn attention for their two main features. First, they enable non-thermal ablation (cold ablation), a processing technique that yields little thermal or secondary damage around the irradiation area [29,30,31]. Second, they can process dielectric materials, such as transparent glass, by nonlinear absorption [32,33,34]. Non-thermal ablation (cold ablation) is possible because it can be processed within the duration of a pulse width shorter than the thermal relaxation time of the material. Metals typically have a thermal relaxation time range of several to tens of picoseconds. In contrast, long pulse or continuous wave lasers use a pulse width with a longer duration than the thermal relaxation time, thus causing a large peripheral heat effect and thermal ablation (hot ablation), which leads to secondary damage such as debris and cracks.

However, due to technological advancements, ultrashort-pulsed lasers no longer produce only non-thermal ablation (cold ablation). The development of ultrashort-pulsed laser technology has progressed to enable a performance at higher power and higher pulse repetition rates. Ultrashort-pulsed lasers initially performed at a few W of power and pulse repetition rates of a few kHz; however, now, they are capable of performing at a power of several hundreds of watts and a pulse repetition rate of several tens of megahertz. As a result of this technological development, the effect of heat can no longer be ignored as heat accumulation is concomitant with a high pulse repetition rate and high power [35,36,37].

Due to the above-mentioned disadvantages, the material characteristics and various laser parameters must be considered together for accurate ultrashort-pulsed laser processing. This is because non-thermal ablation (cold ablation) is no longer overwhelmingly dominant in ultrashort-pulsed laser processing; thermal ablation (hot ablation) may become dominant depending on the material characteristics and laser parameters or thermal (hot) and non-thermal (cold) ablation effects may appear simultaneously. Studies on ablation using ultrashort-pulsed UV wavelength lasers for PET and PI materials, which are materials often used in flexible displays, are still insufficient. To the best of the authors’ knowledge, no study has compared the processing characteristics of materials that show completely different optical characteristics such as PI, which exhibits linear absorption (when the wavelength is 355 nm), and PET, which does not. Moreover, there is no information on their processing characteristics at different laser pulse repetition rates.

Therefore, this study directly compared the processing characteristics of two materials which are commonly used in flexible displays and batteries: PI, which undergoes linear absorption (when the wavelength is 355 nm), and PET, which does not, using an ultrashort-pulsed UV wavelength laser. Different irradiation power conditions, according to the laser pulse repetition rate, were examined to observe the resulting processing characteristics. Then, the processing trends caused by the differences in the absorption characteristics and laser pulse repetition rate of materials were compared and analyzed.

## 2. Materials and Methods

The configuration of the laser processing system used in this study is shown in Figure 1. The polymer samples were placed on a tilting stage, which could be adjusted to an optimum position to avoid unequal vertical displacement over the range of the sample surface from the focal plane during laser irradiation. The tilting stage was then placed on a movable dual axis x-y stage, and checked for no vertical displacement during lateral movement. For our experiments, we used an Nd:YVO_4_ picosecond (ps) pulsed laser with a pulse duration of 10 ps, wavelength of 355 nm, and repetition rate of 50 kHz to 500 kHz (time bandwidth). The pulse repetition rate was set by referring to the specifications provided by the laser manufacturer. In this experiment, the laser beam was irradiated in a direction perpendicular to the sample using an objective lens (Mitutoyo) with 10× magnification, 0.28 numerical aperture (NA), and 20 mm focal distance. When the focal plane was located on the top surface of the sample, the diameter of the laser beam was calculated to be approximately 1.2 μm by the following Equation (1) [38]:(1)$$d=\frac{4M^{2}\lambda f}{\pi D}$$
where d is the diameter of the laser beam on the focal plane, *M*^2^ is the beam quality factor, *λ* is the wavelength, *f* is the focal distance of the focal lens, and *D* is the diameter of the laser beam incident on the lens. The beam quality factor of the laser was 1.3 according to manufacturer’s specifications. The beam that was entered as an objective lens had a diameter of 10 mm.

Laser irradiation was performed under two conditions in this study. The first was a stationary condition in which the laser was irradiated in a fixed position, and the second condition was a scanning condition in which the laser was irradiated while the sample was moved using the y-stage. First, in the stationary beam irradiation condition, the number of incident pulses was used as an experimental variable. The number of pulses was changed from 10 to 15,000 during the experiment (the pulse repetition rate used in the experiment was 50 kHz to 500 kHz). Next, in the scanning beam irradiation condition, the scan speed was used as an experimental variable and was changed from 7 mm/s to 70 mm/s during the experiment (the pulse repetition rate used in the experiment was 50 kHz to 500 kHz). This scan speed was selected through results obtained in previous research. Before changing from one experimental condition to another, the sample was moved at fixed intervals, using the x-stage, to conduct experiments at different positions of the sample.

In this study, the following laser power conditions were applied: the same pulse energy (different average power) and the same average power (different pulse energy). Each of these two conditions is described in detail below.

In the first case, the energy per pulse was the same regardless of a high or low pulse repetition rate. Under this condition, the number of pulses for the same duration at the high pulse repetition rate was higher than that at the low pulse repetition rate. When more pulses are irradiated for the same duration, the intervals between pulses become reduced which results in the material becoming heated again before it has completely cooled. This will cause a heat accumulation effect and increase the collateral damage.

Second, in the case of the same average power, single pulse energy at the high pulse repetition rate is smaller than that at the low pulse repetition rate. Unlike the case of the same pulse energy, single pulse energy is dependent on the pulse repetition rate. As the single pulse energy varies, it is likely to generate collateral damage that is different from that observed with same pulse energy. When the heat accumulation effect becomes dominant due to the short intervals between pulses, the collateral damage area at the high pulse repetition rate is expected to be greater than at the low pulse repetition rate. By contrast, if the magnitude of the single pulse energy is dominant, the opposite trend is anticipated. Thus, under the condition of the same average power, the collateral damage is expected to vary depending on the dominant parameter, single pulse energy or pulse repetition rate.

As discussed above, the morphological characteristics, such as laser-induced ablation and collateral damage, caused by the two conditions of laser power, were compared in this study. For the laser power used in this experiment, a value higher than the damage threshold fluence was selected, based on previous research, to be 0.8 μJ (70 J/cm^2^) for the same pulse energy. Furthermore, the same average power of 0.4 W was applied, for each pulse repetition rate, at a fluence of 700 J/cm^2^ (50 kHz) and 70 J/cm^2^ (500 kHz), respectively.

In the previous study that examined optical waveguide processing with ceramic materials, the criteria for classifying thermal (hot) processing and non-thermal (cold) processing according to the pulse repetition rate and the thermal diffusivity of the material were reported [39]. In this paper, the critical frequency is calculated by the thermal diffusivity and the laser focal spot diameter. The normalized frequency can be calculated from the critical frequency and the laser pulse repetition rate. If the calculated normalized frequency is close to 0, non-thermal (cold) processing is likely to occur during laser processing, and when a normalized frequency greater than 1 is calculated, thermal (hot) processing is likely to occur. At this time, the critical frequency and normalized frequency are calculated by the expression below [39,40].
(2)$$f_{cr}=\frac{D_{th}}{d_{laser}^{2}}\;\;\;\;f_{n}=\frac{f_{laser}}{f_{cr}}$$
where $f_{cr}$ is the critical frequency, $D_{th}$ is the thermal diffusivity, $d_{laser}^{2}$ is the focal spot diameter, $f_{n}$ is the normalized frequency, and $f_{laser}$ is the laser pulsed repetition rate. Based on Equation (2) above, the critical frequency and normalized frequency for PET and PI are calculated with the pulse repetition rate used in our experiment, as follows.

First, in the case of PET, the critical frequency is 60 kHz, and the normalized frequencies for the pulse repetition rates of 50 kHz and 500 kHz are 0.83 (50 kHz) and 8.28 (500 kHz), respectively. In the case of PI, the critical frequency is 54 kHz, and the normalized frequencies for the pulse repetition rates of 50 kHz and 500 kHz are 0.93 (50 kHz) and 9.29 (500 kHz), respectively. Considering only the thermal diffusivity and pulse repetition rate, ignoring the absorption properties and other factors of the material, we can see that in PET and PI, non-thermal processing at low pulse repetition rates (50 kHz) and thermal processing at high pulse repetition rates (500 kHz) have a high possibility. In addition, two types of polymer samples that have completely different absorption characteristics for the laser wavelength were used. These polymer samples are frequently used in flexible displays and batteries. We selected polyimide (PI, Dupont) which exhibits linear absorption for the chosen laser wavelength and polyethylene terephthalate (PET, Bamberger polymers) which does not. The absorbances of these two materials were measured in the wavelength range of 200 to 800 nm using a spectrophotometer and the results are shown in Figure 2a.

To conduct experiments using the above two polymer samples, they were configured as shown in Figure 2b,c. The flatness of the polymer samples used in this study cannot be easily maintained because the materials are flexible. To solve this problem, a glass slide was positioned at the bottom, and a PET film was placed on it to minimize the effect of the slide. The thickness of the bottom PET was 75 μm, and the adhesive thickness was approximately 15 μm. An acrylic adhesive was used. Figure 2b shows the sample configuration with the PI (75 µm thick) at the top. Figure 2c shows a sample configuration with the PET at the top. The top PET was the same sample as the bottom one, but in this configuration, the PET formed the top surface instead of the PI. Furthermore, the PET thickness was set to the greatest thickness of the PI, 75 μm, to study different materials with the same thickness.

The polymer samples, PET and PI, used in this study have completely different characteristics, not only in the absorption of the laser wavelength, but also in the physical properties of the material. To examine them in more detail, PET with no linear absorption and a melting temperature of approximately 200 °C was used. By contrast, PI, with linear absorption, has a completely different characteristic of lacking a defined melting temperature. Thus, the two polymer materials used in this study exhibited completely different physical properties.

The properties of each material used in the experiment are shown in Table 1.

Previous studies have defined the carbonized (graphitized) temperatures for PI, while also showing it to lack a definitive melting temperature, and that carbonization or graphitization is possible by laser energy irradiation [43,44,45]. As such, the PI used in our experiment was also expected to undergo carbonization or graphitization, by laser energy irradiation, and subsequently have altered electrical characteristics. Therefore, to observe the carbonization or graphitization due to laser irradiation, the changes in electrical characteristics were measured using a probe station (MS tech), as shown in Figure 3. The electrical resistance was measured by positioning two probes in alignment within the laser processing line at a distance of less than 1 mm between them, considering their diameters. Electrical characteristics were measured 100 times for each processing condition.

## 3. Results

In our study, processing trends in PI and PET were observed with respect to different laser parameters (laser power, pulse repetition rate, and irradiation method) and absorption at laser wavelengths. After laser processing, the samples were analyzed by field emission scanning electron microscopy (FE-SEM, Hitachi, Tokyo, Japan).

### 3.1. Stationary Beam Irradiation with the Same Pulse Energy

In the first experiment, we analyzed the effect of stationary beam irradiation with the same pulse energy on PET and PI. In this condition, the laser energy per pulse was constant, irrespective of the pulse repetition rate. The obtained results are shown in Figure 4. In this experiment, we maintained the single pulse energy at 0.8 µJ (70 J/cm^2^). Figure 4a,b show the FE-SEM images of laser-irradiated PET without linear absorption and PI with linear absorption.

The processing results in the case of stationary beam irradiation with the same pulse energy in Figure 4 show that irrespective of the linear absorption of the laser wavelength, the ablation size and the collateral damage of the periphery at the high pulse repetition rate were larger than those at the low pulse repetition rate. It is also necessary to examine the effect of polymer type on the laser processing characteristics; these observations are described below.

In the case of PET, it can be seen that collateral damage occurred at the periphery of the processing section during 10-pulse processing, irrespective of the pulse repetition rate. Furthermore, the ablation size and collateral damage at the periphery increased with an increase in the number of irradiated pulses.

As shown in Figure 4b, the processing of PI was considerably different from that of PET at a low pulse repetition rate. In the case of PET, which does not exhibit linear absorption, collateral damage was observed at the periphery even at low pulse repetition rates; in contrast, at a similar pulse repetition rate, PI did not exhibit any collateral damage. This processing trend is similar to non-thermal ablation (cold ablation), which is a form of laser processing that minimizes collateral damage at the periphery. However, at a high pulse repetition rate, collateral damage was observed in both PET and PI. Furthermore, even in PI, which exhibited linear absorption, it could be observed that the ablation size increased with an increase in the number of pulses at a low pulse repetition rate. Similarly, the ablation size and collateral damage at the periphery increased with an increase in the number of pulses at a high pulse repetition rate. In addition, in terms of collateral damage, PET sustained a greater degree of damage at the periphery than PI.

Hence, during laser processing of PI and PET (which exhibit different absorption characteristics) with stationary beam irradiation with the same pulse energy, collateral damage at the periphery was caused by processing at high pulse repetition rates.

### 3.2. Stationary Beam Irradiation with the Same Average Power

We examined the laser processing of PI and PET with stationary beam irradiation at the same average power (using the same irradiation method) but under different power conditions. In this condition, the single pulse energy varies depending on the pulse repetition rate. The single pulse energy is smaller at high pulse repetition rates rather than at low pulse repetition rates. This is a very interesting condition because the collateral damage changes depending on which factor (the single pulse energy or narrow pulse spacing) is dominant. In other words, collateral damage may be larger at a low pulse repetition rate with large single pulse energy or when the pulse repetition rate is high with a narrow pulse spacing.

In our study, we maintained the same average power at 0.4 W, which yielded a fluence of 700 J/cm^2^ at 50 kHz and 70 J/cm^2^ at 500 kHz. Under these conditions, the processing characteristics of PET were as follows. In Figure 5, which shows the processing results for PET, it can be seen that a large amount of collateral damage occurred at 50 kHz, which is the case for a higher single pulse energy, than at 500 kHz. Furthermore, irrespective of the pulse repetition rate, a lot of collateral damage was generated with an increase in the number of pulses. It can also be observed that collateral damage occurred with a very small number of pulses (from 10 pulses).

When the collateral damage at the periphery in Figure 5 is compared with that observed under the same average power condition, it can be noted that the damage was greater at 50 kHz with high single pulse energy than at 500 kHz. In addition, the center of the irradiated laser was processed in a circular shape irrespective of the pulse repetition rate until the number of irradiated pulses was ~100. However, after 5000 pulses, the center of the irradiated laser did not form a circular shape even though the number of pulses increased; instead, it closed again due to re-solidification of the melted material.

Thus, from the results of laser processing of PET under the same average power condition, it can be inferred that collateral damage at the periphery is larger at low pulse repetition rates (high single pulse energy) than at high pulse repetition rates.

We shall discuss the processing of PI, whose laser absorption characteristics are quite different from those of PET, under the same average power condition. From Figure 6, it can be concluded that the processing characteristics of PET and PI were different. The latter exhibited almost no collateral damage at the periphery with less than 1000 pulses at 50 and 500 kHz. A more interesting phenomenon is that completely different morphologies were observed after 5000 pulses at a high pulse repetition rate. This trend, in which there was no collateral damage at the periphery, is similar to the result obtained in the same pulse energy condition at a low repetition rate. (Figure 4b).

When the low pulse repetition rate is considered (Figure 6) in PI, when the number of pulses increased, only the ablation diameter increased (there was no collateral damage at the periphery). This is completely opposite to the result observed with PET under the same average power condition; in this case, the ablation diameter increased along with collateral damage.

In the case of the high pulse repetition rate, up to 1000 pulses, the ablation diameter increased with no collateral damage. However, beyond 5000 pulses, the morphology of the sample changed drastically. When the morphology was examined in detail, the shapes of ablation could be discerned before 1000 pulses, but after 5000 pulses, the ablation rose above the surface of the sample. Many pores were formed on the surface of the raised portion. In addition, during this morphological transformation, the damage sustained at a high pulse repetition rate was greater than that at a low repetition rate when the single pulse energy was greater. This trend could not be observed in PET, in which the damage size increased with an increase in the number of pulses.

### 3.3. Scanning Beam Irradiation with the Same Average Power

In Section 3.1 and Section 3.2, we compared and analyzed two laser power conditions for PI and PET using a stationary beam. The laser power conditions were divided into the same pulse energy and the same average power. The obtained results indicated that the collateral damage at the periphery was larger at the high pulse repetition rate in the same pulse energy condition irrespective of the linear absorption characteristics. However, in case of the same average power, different trends were observed depending on the absorption characteristics of the polymer. Therefore, we chose to study only the same average power condition to conduct experiments with scanning beam irradiation; the fluence used was similar to that in the stationary beam irradiation studies.

Figure 7a shows the results of PET processing under the same average power condition. Collateral damage at the periphery was observed in all samples, irrespective of the scan speed and pulse repetition rate. A larger collateral damage was observed at the low pulse repetition rate than at the high repetition rate with high single pulse energy. This is similar to the results observed for stationary beam irradiation with the same average power. In the stationary beam irradiation condition, greater collateral damage occurred at a low repetition rate with higher single pulse energy than at a high repetition rate. Furthermore, at a given pulse repetition rate, collateral damage at the periphery was greater when the scan speed was slower.

The results of PI processing at same average power condition are shown in Figure 7b. When the material was processed at a high repetition rate and slow scan speed (7 mm/s), the obtained morphology was completely different from that observed under other conditions. In this case, the surface turned porous; furthermore, a part of the processed surface peeled off and rose above the normal sample surface. In addition, the damage was greater than that at a low repetition rate. This result is similar to that observed in Figure 6, corresponding to stationary beam irradiation. In Figure 6, a general shape of ablation can be observed when the number of pulses at a high repetition rate was smaller than 5000; beyond this point, drastic changes occurred in the morphology of the sample.

When PI and PET were processed by scanning beam irradiation with the same average power, collateral damage was observed under all conditions for PET and no peculiar phenomenon could be noted. In contrast, a morphological change was observed in PI when it was processed at a high pulse repetition rate and slow scan speed; furthermore, greater damage could be observed at a low pulse repetition rate.

The images obtained for the PI sample using a reflective optical microscope when the laser parameters used caused a morphological change during processing by scanning beam irradiation with the same average power condition are examined. The images obtained from the samples for the case of morphological changes, which were observed in the FE-SEM images using a reflective optical microscope, and for another case, are shown in Figure 8. From the FE-SEM images, it can be observed that the scan line irradiated by the laser changed to black at the high pulse repetition rate, where the sample exhibited a porous surface structure and a rising shape above the surface. In contrast, no black area along the scan line was observed at the low pulse repetition rate, where the sample exhibited the general shape of laser-induced damage.

Thus, clearly contradictive results can be observed in the reflective optical microscope images between the two processes of low and high pulse repetition rates, indicating clear morphological differences in the FE-SEM images. Next, the electrical characteristics measurement results will be discussed to verify whether the blackened region is caused by carbonization or graphitization.

### 3.4. Electrical Resistance Results Obtained for a PI Sample Using Scanning Beam Irradiation

Previously, we discussed the SEM and optical microscopy observations corresponding to PET and PI processed using different laser parameters. In this section, measurements were conducted to verify whether the blackened scan line that was irradiated by the laser and the porous surface structure in PI corresponded to carbonization. To evaluate these changes, the electrical characteristics of laser-processed PI were measured using a probe station, as shown in Figure 3. The probes were placed at two points on the laser-irradiated scan line and the electrical resistance of the sample was measured.

We did not measure the electrical resistance of PET because it did not exhibit morphological change, which is assumed to be a result of carbonization or graphitization. Furthermore, the electrical resistance of only those PI samples processed by a scanning beam was measured (Figure 9). The electrical resistance of the PI sample without any processing was several GΩ. The measurement results of the laser-processed sample (in the case of a high scan speed (70 mm/s)) indicated almost no difference compared to the results before laser processing for each pulse repetition rate. However, at a slow scan speed (7 mm/s), completely different results were observed depending on the pulse repetition rate. Although the resistance at low repetition rates (50 and 100 kHz) was not very different from that of the unprocessed sample, at high pulse repetition rates (200 and 500 kHz), a significant difference could be observed. In this case, it can be seen that the electrical resistance of the processed samples decreased significantly.

The standard deviation is 11.53 and 14.85, respectively, for 200 kHz and 500 kHz, where the carbonization occurred.

In summary, when PI, which exhibited linear absorption, was processed using a scanning beam with the same average power (slow scan speed (7 mm/s) and a high pulse repetition rate), morphological changes were observed and the electrical resistance decreased significantly.

## 4. Discussion

Generally, the laser processing characteristics are affected by various parameters. Among these, the following three characteristics are examined in particular: first, the correlation between the material’s thermal relaxation time and laser pulse width; second, the occurrence of the heat accumulation effect due to the pulse repetition rate; and third, the difference in the absorption mechanism of the laser photon energy inside the material. Each of these characteristics is explained in detail below.

The first characteristic is the correlation between the thermal relaxation time and laser pulse width. Thermal relaxation time represents how fast the kinetic energy (thermal energy) of electrons that absorbed laser energy is transferred to the surrounding lattice.

The thermal relaxation time varies by material type, and the thermal relaxation times of metals range from 1 to 10 ps, whereas those of nonmetals range from several tens to hundreds of picoseconds. If the laser pulse width used in laser processing is shorter than the thermal relaxation time of the material, it is thermally confined and laser processing is performed even before thermal energy is transferred to the surrounding lattice; this enables non-thermal ablation (cold ablation) with a small amount of collateral damage in the periphery. Thus, the laser used in this study (pulse width = 10 ps) was expected to enable non-thermal ablation (cold ablation) when only the thermal relaxation times of PI and PET are considered.

The second characteristic is the occurrence of heat accumulation depending on the pulse repetition rate. Heat accumulation refers to the phenomenon in which the base temperature increases around the spot where the laser is irradiated; this is because thermal energy is accumulated due to continuous incident pulses before the heat induced by previous pulses can be cooled down sufficiently. Therefore, more residual heat may be generated using high single pulse energy or with a large number of repeated pulses at a high repetition rate.

To examine the occurrence of heat accumulation at a pulse width of 10 ps, we measured the thermal distribution in PI samples subjected to scanning beam irradiation using a high-speed infrared (IR) camera (TELOPS; 2000 fps). In this case, the fluence and scan speed were 700 J/cm^2^ and 70 mm/s, respectively. The obtained results are shown in Figure 10. In this figure, the top part represents the time when scan processing started and the bottom part is the thermal distribution when the scan process was almost complete. The pink circle represents the part heated by laser irradiation and the surrounding blue circle on the left side shows the result of preheating during laser scan processing. In addition, the part that looks like a comet tail shows the distribution of the residual heat generated by laser irradiation. Thus, it could be confirmed that direct heating due to laser irradiation generated heat, preheating in front of the laser irradiation part, and residual heat behind it. Hence, it is possible that thermal ablation (hot ablation) occurred instead of non-thermal ablation (cold ablation) even though we used an ultrashort-pulse laser.

The third characteristic is the difference in the absorption mechanism of the laser photon energy inside the material. Here, absorption is a phenomenon in which energy (a light source) is absorbed into a material and converted to a different energy level. These absorptions can be classified into linear absorption and nonlinear absorption [46]. Linear absorption can be subdivided again into thermal absorption that is caused by the transfer between the vibration and rotation levels of electrons, and photochemical absorption which occurs when photon energy above the molecular binding energy is irradiated and the bonds of molecules are broken [46].

In the case of the PI material used in this study, the existence of linear absorption at the laser wavelength can be confirmed by referring to Figure 2a, which shows the measurement result of the spectrophotometer. Between the two types of linear absorption, photochemical absorption is considered to be more dominant in PI materials. Thus, the discussion of PI will now focus on photochemical absorption.

To investigate the photochemical absorption of the PI material, the chemical structure was examined to verify the binding energy of molecules comprising the PI material, and the C-N bond was observed. According to previous studies, the binding energy for this C-N bond is approximately 3.04 eV [38]. The photon energy of the laser wavelength used here is calculated to be approximately 3.49 eV. Hence, the laser photon energy of the PI material is greater than the C-N binding energy. In other words, photochemical absorption will occur in the PI material because the laser photon energy is larger than the binding energy of molecules.

When laser processing is performed in conditions in which photochemical absorption, such as that in PI, is possible, non-thermal ablation (cold ablation) might occur and this minimizes collateral damage at the periphery of the laser-irradiated area (this is because non-thermal processing might be carried out).

Meanwhile, nonlinear absorption refers to absorption that depends on the intensity of the laser source (light source). In this case, even when a laser of the same wavelength is used, if the intensity of the laser is weak, energy is transmitted without being absorbed by the material. In contrast, if the laser intensity is high, energy is absorbed. Such nonlinear absorption may be attributed to nonlinear phenomena such as multiphoton absorption and avalanche ionization when the laser pulse width is of the order of several picoseconds or if the laser intensity is higher than 10^13^ W/cm^2^ [46].

When collateral damage is considered in the ultrashort-pulsed laser processing, as in our study (occurrence of non-thermal (cold) or thermal (hot) ablation), the case of processing a transparent material by nonlinear absorption and the case of processing a material in which linear absorption occurs (considering only the relationship between pulse width and the material’s thermal relaxation time) must be examined separately. To explain in more detail, when the thermal relaxation time of the material is shorter than the pulse width in a process using an ultrashort-pulsed laser, non-thermal (cold ablation), which minimizes the collateral damage in the periphery of the laser-irradiated area, is certainly possible. However, if a material without linear absorption at the laser wavelength is processed, nonlinear absorption is dominant during the process, and it is difficult to predict whether processing close to non-thermal ablation (cold ablation), which minimizes collateral damage in the periphery of the laser-irradiated area, will be possible or if it will be close to thermal ablation (hot ablation), which generates collateral damage.

Using this background, we shall discuss the processing of PET. In Figure 2a, which shows the spectrophotometry results of the PET material used in our study, no linear absorption could be observed (at the laser wavelength). In the case of this nonlinear absorption material, we do not know whether non-thermal ablation (cold ablation) or thermal ablation (hot ablation) occurs. However, if we consider only the relationship between the material’s thermal relaxation time and the pulse width of the ultrashort pulse laser, it is expected that non-thermal ablation (cold ablation) might occur, as the pulse width was only 10 ps, which is shorter than the material’s thermal relaxation time.

We shall consider the case of PI and PET, which exhibit linear and nonlinear absorption, respectively, from the viewpoint of other parameters that affect laser processing characteristics.

In most of the previous studies on PET processing using ultrashort-pulse lasers, it was reported that non-thermal ablation (cold ablation) occurred. Our team also investigated the micromachining of PET materials using UV picosecond lasers [47]. In that study, a UV picosecond laser (same repetition rate of 50 kHz), with specifications similar to those in this study, was used. The fluence range used was 5–30 J/cm^2^, which was at least 10 times smaller than the fluence range used in the current investigation. In our past study, non-thermal ablation (cold ablation), with minimal collateral damage at the periphery of the laser-induced crater, was observed in PET, similar to the results reported by other researchers.

Based on the cited results, it was predicted that processing close to non-thermal ablation (cold ablation) would be possible with the ultrashort-pulse laser used in this study [47]. However, the laser processing results in the present work indicated a trend close to thermal ablation (hot ablation), where collateral damage accumulates in the periphery of the laser-induced crater. This is different from our predictions regarding the laser parameters (fluence, laser pulse repetition rate, and laser beam irradiation method) used in this study.

The specific details regarding this processing trend will be described separately for stationary beam irradiation (the same pulse energy case and the same power case) and scanning beam irradiation (the same power case).

First, we examined the case of stationary beam irradiation with the same pulse energy. In this instance, PET exhibited collateral damage at the periphery of the laser-induced crater, irrespective of the parameters (pulse repetition rate and the number of pulses). In other words, thermal ablation (hot ablation) occurred instead of non-thermal ablation (cold ablation). A closer look indicated that thermal ablation (hot ablation) occurred in the chosen fluence range even when only a few pulses were used. It should be noted that in this study, even when the pulse width of the laser was shorter than the thermal relaxation time of the material, heat accumulation occurred, resulting in thermal ablation (hot ablation) rather than non-thermal ablation (cold ablation).

In addition, a larger amount of collateral damage was observed at the periphery at high pulse repetition rates with a very narrow spacing between the pulses than at low repetition rates under the same pulse energy condition. This difference might be attributed to greater heat accumulation at high pulse repetition rates.

Second, the results of laser processing using stationary beam irradiation with the same average power are examined. In this condition as well, collateral damage occurred in every case, regardless of the pulse repetition rate and number of pulses. Here, too, thermal ablation (hot ablation) occurred instead of non-thermal ablation (cold ablation). However, there were some differences when compared to the same pulse energy condition. At the same average power, more collateral damage occurred at low pulse repetition rates than at high repetition rates.

To examine this result in detail, we can note the difference in the single pulse energy and pulse spacing between high and low repetition rates. When there was a 10-fold difference between the two rates, at the low pulse repetition rate, the single pulse energy was 10 times higher, but the pulse spacing was 10 times wider. When the collateral damage is considered, a large amount of collateral damage occurred naturally at higher single pulse energies. However, due to heat accumulation, a narrow pulse spacing may lead to a larger amount of collateral damage.

Based on this discussion, the collateral damage generated at high and low repetition rates at the same average power can be compared. When only the effect of the single pulse energy is considered, a low pulse repetition rate causes large collateral damage. However, if only pulse spacing is considered, aggressive heat accumulation can occur at a high pulse repetition rate, thus causing greater collateral damage.

In the case of PET without linear absorption, a larger amount of collateral damage occurred at a low pulse repetition rate with high single pulse energy than that at the high repetition rate with a narrow pulse spacing. This result suggests that in PET, the effect of the level of single pulse energy on the collateral damage was dominant.

Third, the results of laser processing using scanning beam irradiation with the same average power were examined. Irrespective of the process parameters (scan speed and repetition rate), thermal ablation (hot ablation) was observed along with collateral damage. Furthermore, greater collateral damage occurred at low pulse repetition rates than at high repetition rates.

In the case of PET, the effect of the single pulse energy was more dominant than that of pulse spacing even when it was processed by scanning beam irradiation. This is considered to be the reason for the greater collateral damage observed at low pulse repetition rates than at high repetition rates.

In terms of scan speed, processing at a slow scan speed generates greater collateral damage than that at a fast scan speed. If the scan speed is small, the moving distance for the sample stage in a given time period is reduced and the laser energy irradiated at one location increases, leading to heat accumulation.

When we examine the processing results using various laser parameters with PET without linear absorption, the processing trend of thermal ablation (hot ablation), which generates collateral damage on the periphery, was observed for every parameter. This processing trend is opposite to the result of our previous study in which we used the same laser, and the ultrashort-pulsed laser processing results of other researchers, which showed that non-thermal ablation (cold ablation) was possible. The reason that this opposite result was observed seems to be that a larger amount of collateral damage occurred because the fluence range was at least 10 times higher than that in the previous studies.

Furthermore, it can be thought that the thermal effect hardly occurs because the pulse width of the laser that we used was shorter than the thermal relaxation time of the PET material. However, as shown in the above result of measuring the thermal distribution using a high-speed IR camera in Figure 10, even an ultrashort-pulsed laser is not 100% free from the thermal effect. As shown in the measurement result, minimal thermal effects are observed (pre-heating and residual heat). The size of these thermal effects will naturally increase with the fluence.

Putting these observations together, in previous studies, laser processing was conducted in a non-thermal ablation (cold ablation) window, whereas we performed laser processing in a thermal ablation (hot ablation) window. Therefore, the present study showed processing trends that changed from non-thermal (cold) to thermal (hot) ablation with the chosen laser parameters, even though PET without linear absorption was processed using an ultrashort UV pulsed laser. Consequently, laser parameters need to be designed after considering thermal effects, such as collateral damage, to precisely conduct processing on PET with ultrashort-pulse lasers.

Before described the PI results, with possible linear absorption (photochemical absorption), the main results of PET processing will be summarized here. As the laser used in this study was shorter than the material’s thermal relaxation time, we expected a non-thermal ablation (cold ablation) window. However, contrary to our expectations, a thermal (hot) ablation window was observed due to the high fluence.

The PI material, with linear absorption (photochemical absorption), can also cause a non-thermal ablation (cold ablation) window or a thermal ablation (hot ablation) window, if only the laser pulse width is considered. Before discussing the results of the processing trend for the PI material, the differences in the absorption characteristics between the PET and PI samples should be noted. The laser energy absorption by the PET sample starts as nonlinear absorption because the sample does not exhibit linear absorption. However, in the case of the PI sample, which exhibits linear absorption, the laser energy absorption starts as photochemical absorption, which is a type of linear absorption. When laser processing is performed under a condition that enables this photochemical absorption, it enables selective ablation because collateral damage in the periphery can be minimized. When we consider only this absorption characteristic, PI presents more amenable conditions for non-thermal ablation (cold ablation). This aspect will be discussed in more detail in the rest of this section.

The results for the PI sample, which was processed with a 10-ps pulsed laser of a UV wavelength and for which photochemical absorption is possible, can be largely summarized in two parts. The first is the processing result in the non-thermal ablation (cold ablation) window and the second is the processing result in the thermal ablation (hot ablation) window.

In non-thermal ablation (cold ablation) window, the laser processing parameters were as follows: at a low repetition rate, minimal collateral damage could be observed at the periphery of the laser-induced crater irrespective of the laser irradiation method (stationary beam or scanning beam) or laser power condition (same pulse energy or same average power). At a high pulse repetition rate, processing with minimal collateral damage was possible only under some conditions. In the stationary beam irradiation condition, it was possible to minimize collateral damage when the number of irradiated pulses was small (less than 1000). In the scanning beam irradiation condition, collateral damage could be reduced at fast scanning speeds (70 mm/s). Using SEM, features corresponding to a non-thermal ablation (cold ablation) window could be inferred.

Later, processing in the thermal ablation (hot ablation) window and the phenomenon of carbonization was examined. Thermal ablation (hot ablation) occurred under a high pulse repetition rate condition. In the case of stationary beam irradiation with the same pulse energy, collateral damage was generated regardless of the number of pulses. In the case of stationary beam irradiation with the same average power and different laser power conditions, collateral damage could be observed when the number of irradiated pulses was large (5000 or more). Furthermore, with scanning beam irradiation and a high repetition rate, collateral damage occurred at slow scan speeds (7 mm/s). When the material was processed at a high pulse repetition rate with a narrow pulse spacing at the same pulse energy, collateral damage occurred at the periphery of the laser-induced crater due to heat accumulation. In this phenomenon, thermal energy is accumulated due to continuous pulses before the thermal energy induced by previous pulses can be sufficiently dissipated.

The results obtained with the same average power are examined separately for stationary beam and scanning beam irradiation. In the case of stationary beam irradiation, when many pulses (5000 or more) are irradiated at a high pulse repetition rate, non-thermal ablation (cold ablation) converts into thermal ablation (hot ablation). Meanwhile, in scanning beam irradiation, when the material is processed at a high pulse repetition rate and slow scan speed, once again, non-thermal ablation (cold ablation) converts into thermal ablation (hot ablation). These two cases are explained in more detail below.

In the first case of stationary beam irradiation, when the number of irradiated pulses increases (5000 or more) at a high pulse repetition rate, morphological characteristics that were completely different from those observed at a small number of pulses could be observed. In the latter case, non-thermal ablation (cold ablation) caused minimal collateral damage (small number of pulses). However, when the number of pulses increased to 5000, the surface of the sample rose like a volcano, which transformed the porous structure of the sample. When the number of pulses increased after this morphological change, the processed diameter and the number of porous structures on the surface also tended to increase.

The morphological changes examined so far are interesting because the laser-processed diameter by the single pulse energy and collateral damage in the periphery of the laser-induced crater must be large at a low pulse repetition rate, considering only the single pulse energy for the laser power conditions used. However, for the PI sample with linear absorption, the processed diameter and collateral damage in the periphery of the laser-induced crater were larger at a high pulse repetition rate with low single pulse energy and narrow pulse spacing.

In the second case of scanning beam irradiation, a morphological change occurred when the material was processed at a slow scan speed (7 mm/s) at the high pulse repetition rate. The form of morphological change observed was similar to that observed in stationary beam irradiation. At a slow scan speed (7 mm/s) at the low pulse repetition rate, collateral damage did not particularly expand close to the laser scan line. However, at high pulse repetition rates (slow scan speed), a peel-off shape with a porous structure could be observed along the laser scan line. When samples processed at low and high pulse repetition rates were compared (Figure 8), stark differences were noted. In the case of the high pulse repetition rate sample in which morphological changes occurred after scanning beam irradiation, a black carbonization region was observed in the laser-irradiated area. In contrast, no such area could be observed at low pulse repetition rates.

The PI samples processed by stationary and scanning beam irradiation were observed using FE-SEM and reflective optical microscope images. They exhibited thermal ablation (hot ablation) zones when the number of pulses was large (5000 or more) or at slow scan speeds (7 mm/s).

The morphological characteristics, such as the porous surface structure and black region due to the occurrence of thermal ablation (hot ablation), in the PI sample processing results observed in this work are unique features of carbonization. This porous carbonization has also been reported in previous studies [43,48]. The results obtained from these studies were very similar to those obtained from our study using SEM and photo images. The SEM images of the carbonized material surface showed porous surface structures, which were raised above the surface, similar to the results of our study. Moreover, the photo image shows that the carbonized surface turned black along the laser-irradiated line. When carbonization occurs, a change in the electrical characteristics of the PI sample can be observed (insulator material -> conductive material). Additionally, the electrical resistance caused by the carbonization of the PI sample’s surface varies with respect to the laser parameters (power and scan speed).

As discussed above, the electrical characteristics of the samples were measured using a probe station, as shown in Figure 3, to verify whether the blackening and porous surface structures observed in this study exhibit electrical characteristics similar to those obtained from the carbonization or graphitization results reported in previous studies.

The measurement results regarding the variation in the electrical characteristics of the samples, which were laser processed using a probe station, are shown in Figure 9. These results show that the electrical resistance measured was low at high pulse repetition rates (200 kHz and 500 kHz), where the porous surface structure and peel-off shape were observed. This value is 1/10^7^ of that of the electrical resistance before material processing. In other words, the PI material was an insulator before laser processing, but its conductivity increased by carbonization or graphitization after laser processing.

In contrast, in the case of the laser processing results using low pulse repetition rates (50 kHz and 100 kHz), which resulted in a generic shape of laser-induced damage due to laser processing, the electrical resistance measured using the probe station was very high. This result is not very different from the electrical resistance of the PI material before laser processing. In other words, carbonization or graphitization did not occur. Therefore, the sample, which showed a black porous surface structure morphology in the FE-SEM results and reflective optical microscope images, is considered to be carbonized or graphitized, based on the measurement results obtained using the probe station.

When the reason for carbonization or graphitization is examined, it seems to be connected with the physical properties of PI. PI does not have a specific melting temperature, but undergoes carbonization at ~800 °C [49]. When laser processing is conducted at a high pulse repetition rate on PI, heat accumulation cannot be ignored. Thus, as the temperature does not increase beyond the carbonization temperature before certain conditions are met (stationary beam with fewer then 5000 pulses and scanning beam at 70 mm/s), non-thermal ablation (cold ablation) with minimal collateral damage occurs. However, heat accumulation might increase the temperature required for carbonization.

We expected that non-thermal processing would be possible due to photochemical absorption in PI with a 10 ps UV wavelength. However, although non-thermal (cold ablation) was possible at a low pulse repetition rate, thermal ablation (hot ablation) occurred at high pulse repetition rates owing to heat accumulation. Therefore, as all problems cannot be solved by ultrashort-pulse laser processing, it is necessary to optimize the processing parameters.

In this study, we compared and analyzed laser processing of PI and PET, which exhibit different absorption characteristics, in terms of the pulse repetition rate, laser power condition, and laser irradiation method. Next, we calculated the temperature behavior of residual heat in PI in terms of pulse repetition rate using a heat conduction equation and examined the calculated results.

Two numerical models for the analysis of overlapping temperature behavior due to the pulse repetition rate are available. These are the finite difference model and the finite element model. Between them, the finite difference model is advantageous as it offers very effective calculation when only a simple temperature behavior is considered. Thus, the following simplified version of the finite difference model was used.

The temperature distribution, which was caused by residual heat generated by the pulse repetition rate during laser processing, was calculated using the heat conduction equation proposed in previous work. The residual heat generated by the laser pulse energy is defined as follows [35,50]:(3)$$Q_{Heat}=\eta_{Abs}\times \left(1-\eta_{th}\right)\times E_{pulse}=\eta_{Heat}\times E_{pulse}$$
where *Q**_Heat_* is the residual heat, *η**_Abs_* is the absorption rate, *η**_th_* is the thermal efficiency, *E_pulse_* is the pulse energy, and *η**_Heat_* is the ratio of the residual heat converted by the incident pulse energy. The equation for calculating the temperature distribution by multiple pulses at the point source is given as follows [35,50]:(4)$$\Delta T\left(t\right)=\frac{2Q_{Heat}}{\rho \times C_{p}\times \sqrt{{\left(4\pi k\right)}^{3}}}\overset{N_{p}}{\underset{N=1}{\sum }}\frac{\phi \left(t-\frac{N-1}{f_{L}}\right)}{\sqrt{{\left(t-\frac{N-1}{f_{L}}\right)}^{3}}}e^{-\frac{1}{\left(t-\frac{N-1}{f_{L}}\right)}\times \frac{r^{2}}{4k}}$$
where ρ is the density, $c_{p}$ is the specific heat, *k* is the temperature conductivity, *t* is time, $r^{2}=x^{2}+y^{2}+z^{2}$, *N* is the number of pulses, *f_L_* is the pulse repetition rate, and ϕ is the Heaviside function, which is 0 if the value is smaller than 0 and 1 if the value is equal to or larger than 0.

The temperature distribution with respect to the number of incident laser pulses during pulsed laser modeling (where the pulse repetition rate is *f_L_* with a point source) is described in Equation (3).

This equation shows that the temperature increased by the *N*th pulse after a delay of time equal to (*N* − 1)/*f_L_* after the first pulse enters *t* = 0. In addition, to consider temperature distribution due to residual heat in the material, we used the parameters x = y = z = 0. Based on this discussion, it can be inferred that temperature distribution is governed by the relationship between *Q_Heat_*, which is the heat generation term of the laser, the pulse repetition rate of the laser, and the number of continuous pulses.

We used Equations (3) and (4) to analyze temperature behavior due to residual heat in PI subjected to laser processing at high and low pulse repetition rates. In this case, *η*_Heat_ was assumed to be a constant. The obtained results are shown in Figure 11. In this figure, the x-axis represents time and the y-axis represents temperature; the blue line indicates the results related to the low pulse repetition rate while orange lines indicate the results at a high pulse repetition rate. Meanwhile, the gray dashed line indicates the carbonization temperature of PI. In Figure 11a, when one laser pulse is irradiated, the surface temperature increased due to interaction between the material and laser. The heat generated is dissipated by thermal conduction in the material during the dwell time between pulses leading to cooling in the irradiation area. This cooling continues until the next pulse is irradiated. When the next pulse is irradiated during cooling, the temperature increases once again. The lowest temperature before the temperature is raised by the next pulse is plotted by the spiked line. At a low pulse repetition rate, five spikes appeared, but this number increased to 50 at a high pulse repetition rate. This number corresponds to the number of pulses used; the obtained spike-point temperature is called the baseline temperature, which maps the rise in temperature (residual heat) with respect to the number of repeated pulses. The plot was re-drawn as shown in Figure 11b using the baseline temperature and extended to include pulses, 500 pulses in the case of the low repetition rate and 5000 pulses in the case of the high pulse repetition rate.

To understand the basic phenomena corresponding to the temperature behavior related to heat accumulation during the laser processing of PI (as observed in the experimental results), the calculation was simplified by assuming that *η_Heat_* is a constant [35]. We estimated the *η_Heat_* value based on the number of pulses at which carbonization occurred with a high pulse repetition. During stationary beam processing, carbonization occurred after 5000 pulses. In the case of scanning beam irradiation, the number of pulses overlapped by the correlation between the beam size and pulse repetition rate during scan speed processing was approximately 500. Thus, we selected a *η_Heat_* value that makes the baseline temperature rise above the carbonization temperature at 3000 pulses, which is larger than the average of 500 and 5000 pulses.

In Figure 11b, the result value naturally rises above the carbonization temperature because the *η_Heat_* value selected was such that the baseline temperature was greater than the carbonization temperature (based on the experimental results with a high pulse repetition rate). At the same *η_Heat_* value, with the low pulse repetition rate, the baseline temperature was much lower than the carbonization temperature. At the same average power, when only the pulse energy was considered, the calculated baseline temperature at the low pulse repetition rate was 10 times larger than that at the high repetition rate. Thus, it is likely that the highest temperature for the single pulse would be higher in the former case. However, as the dwell time at the low pulse repetition rate is longer than at the high repetition rate, the concomitant cooling duration is also 10 times longer. Therefore, the rate of increase in the baseline temperature is gentler at the low pulse repetition rate than at the high repetition rate. Furthermore, this low baseline temperature corresponds with the occurrence of the non-thermal (cold ablation) observed experimentally. However, in the case of the high pulse repetition rate, thermal ablation (hot ablation) occurred due to the large amount of residual heat generated by heat accumulation.

All the calculations described above using Equations (3) and (4) are based on highly conservative assumptions. Hence, it is possibile that the differences in temperature behaviors at high and low pulse repetition rates are actually greater. However, the results presented here help us understand the basic phenomena, such as residual heat accumulation and the consequent temperature behavior.

## 5. Conclusions

In this study, the processing characteristics of ultrashort-pulse lasers on PET and PI, which exhibit different absorption characteristics, were compared at UV laser wavelengths mainly used in flexible displays and batteries. In addition, material processing characteristics based on the irradiation power at different laser pulse repetition rates were compared. The pulse repetition rate was varied in the range of 50–500 kHz while the fluence was set at 70–700 J/cm^2^. Both stationary beam and scanning beam irradiation were carried out. For the former, the number of pulses was varied from 10 to 15,000, while for the scanning beam, scan speeds of 7 and 70 mm/s were used. Furthermore, two laser power conditions were used—the same pulse energy and the same average power. Residual heat due to the heat accumulation effect is likely to occur when the single pulse energy is large or the pulse repetition rate is high. At the same average power, high single pulse energy and wide spacing between pulses were observed at low pulse repetition rates. In contrast, the high pulse repetition rate included low single pulse energy and narrow spacing between pulses. For this reason, it occurs at a low pulse repetition rate if the residual heat due to heat accumulation or the collateral damage is dominated by the single pulse energy, or at a high pulse repetition rate if it is dominated by the effect of the pulse and pulse spacing.

In PET, which does not exhibit linear absorption, the results were close to thermal (hot ablation) in all laser parameters. In other words, collateral damage accumulated around the laser irradiation area. At the same average power, larger amounts of collateral damage occurred at low pulse repetition rates than at a high pulse repetition rate (10 times higher single pulse energy at low repetition rates).

In contrast, PI, which exhibited linear absorption, experienced a non-thermal ablation (cold ablation) window, a thermal ablation (hot ablation) window, and carbonization. The non-thermal ablation (cold ablation) window occurred in all processing conditions at low pulse repetition rates, with a small pulse number at high repetition rates, and at fast scan speeds. In this case, due to photochemical absorption, non-thermal ablation (cold ablation) with minimal collateral damage occurred. A thermal ablation (hot ablation) window and carbonization (graphite formation) was observed at a high pulse repetition rate in the same pulse energy condition. In the same average power condition, thermal ablation (hot ablation) occurred with a large number of pulses at a high repetition rate or at a slow scan speed. At high pulse repetition rates, collateral damage was generated at the periphery of the laser-induced crater due to heat accumulation. In particular, when the material was processed at the same average power, the collateral damage at the high pulse repetition rate was greater than at the low pulse repetition rate with larger single pulse energy due to morphological changes beyond certain conditions (stationary beam: after 5000 pulses; scanning beam: 7 mm/s). The morphological changes (shape and carbonization) observed in this study for laser-processed PI were similar to those reported earlier. Such morphological changes also represented changes in the electrical resistance of the material and therefore might be attributed to carbonization.

Furthermore, simulations were conducted using the heat conduction equation to analyze baseline temperature behavior during the laser processing of PI. At a low pulse repetition rate, the baseline temperature was not higher than the carbonization temperature, but at high repetition rates, it was higher than the carbonization temperature. This difference might be attributed to heat accumulation. Our experimental results may be utilized as basic data for processing similar materials using ultrashort-pulse UV lasers. In the case of PET, which does not exhibit linear absorption, appropriate laser processing parameters need to be designed. This is because the processing trend is heavily dependent on the laser parameters; in this study, we observed thermal damage in all conditions even though non-thermal ablation (cold ablation) was reported in previous studies. In the case of PI, with linear absorption characteristics, precise processing is possible because it enables non-thermal ablation (cold ablation) at a low pulse repetition rate, whereas at high pulse repetition rates, it undergoes thermal ablation (hot ablation) due to the heat accumulation effect. Therefore, precise parameters should be designed for this case as well.

## Figures and Tables

**Figure 1 polymers-12-02018-f001:**
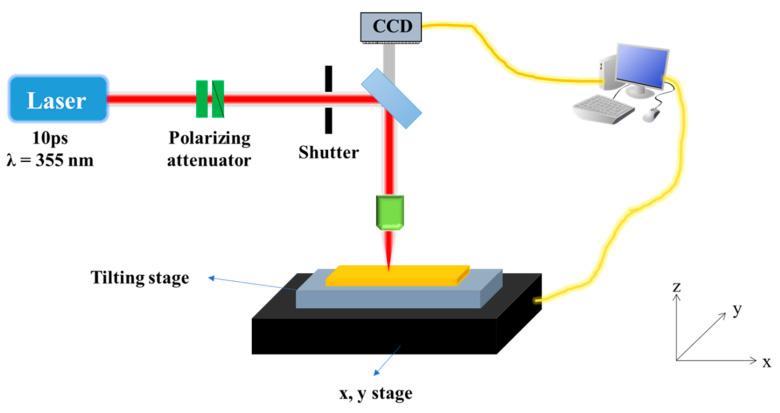
Schematic diagram of a laser processing system for a polymer material.

**Figure 2 polymers-12-02018-f002:**
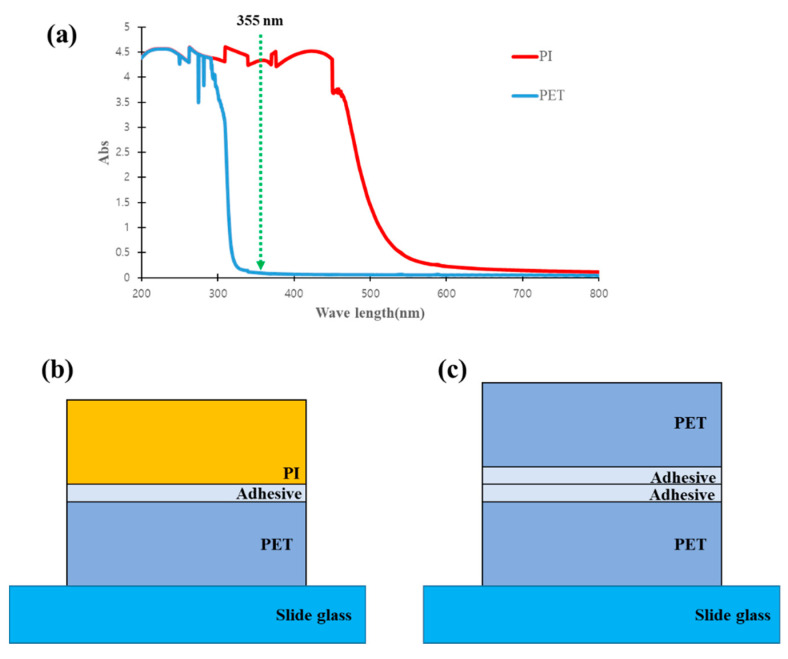
(**a**) Absorption spectrum results for two types of polymer materials using UV–Vis spectrophotometry and configuration of the samples prepared for the experiment: (**b**) polyimide (PI) and (**c**) polyethylene terephthalate (PET).

**Figure 3 polymers-12-02018-f003:**
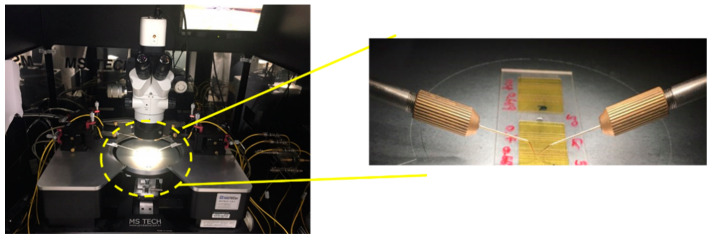
Photo image of the probe station equipment for measuring electrical characteristics.

**Figure 4 polymers-12-02018-f004:**
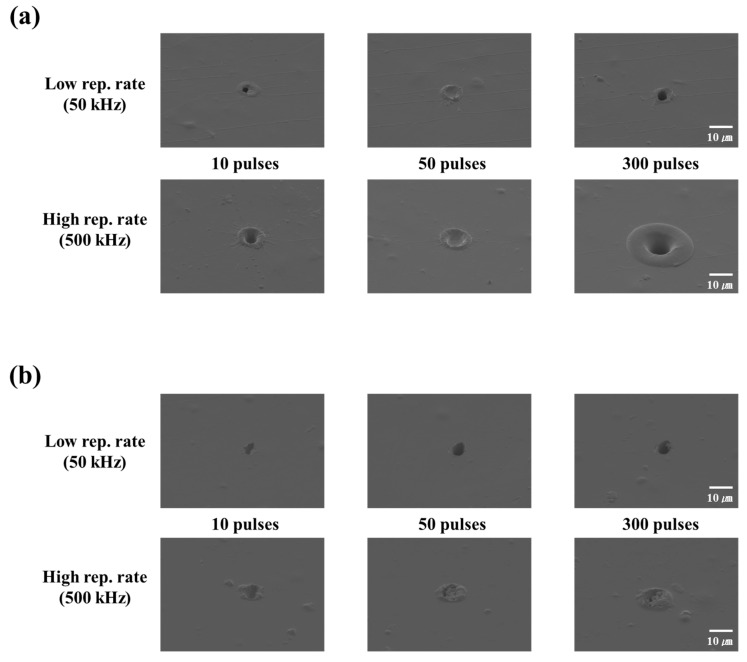
SEM images of the laser processing results under stationary beam irradiation with the same pulse energy (0.8 μJ; 70 J/cm^2^). (**a**) Polyethylene terephthalate (PET) and (**b**) polyimide (PI).

**Figure 5 polymers-12-02018-f005:**
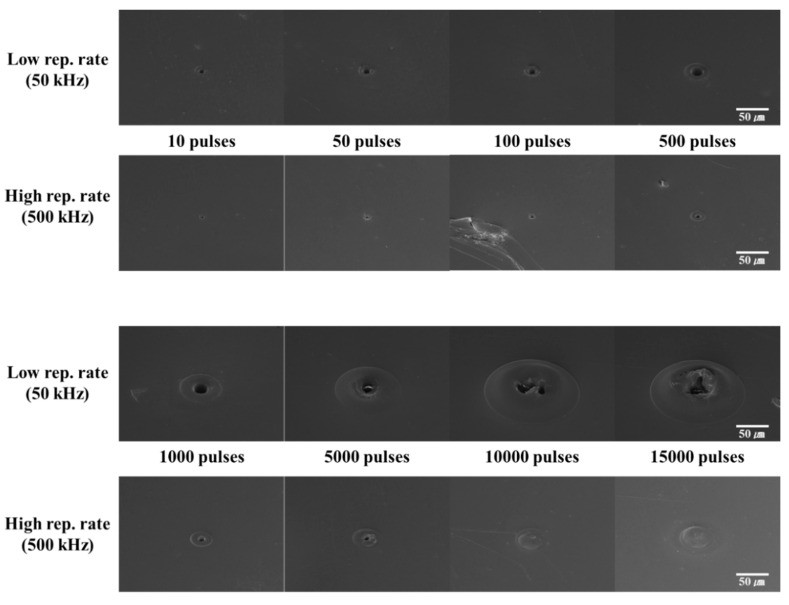
SEM images of the laser processing results of polyethylene terephthalate (PET) under stationary beam irradiation with the same average power.

**Figure 6 polymers-12-02018-f006:**
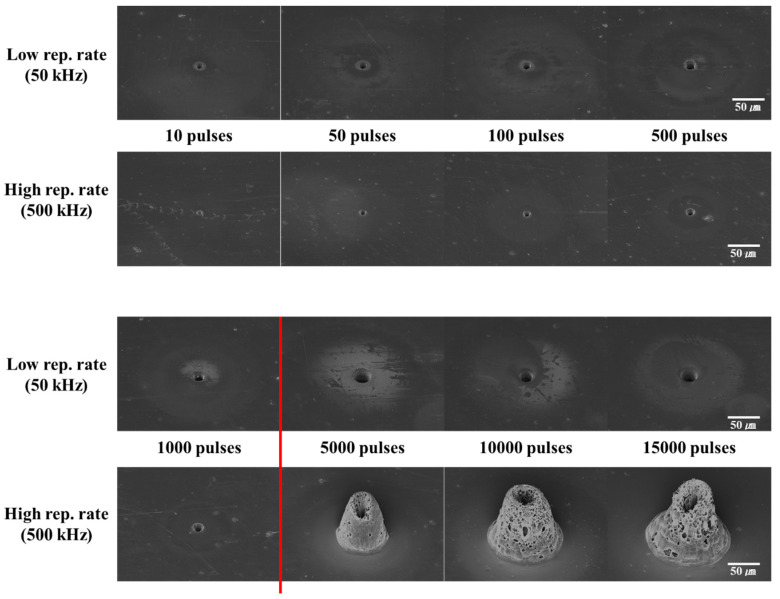
SEM images of laser processing results of polyimide (PI) under stationary beam irradiation with the same average power.

**Figure 7 polymers-12-02018-f007:**
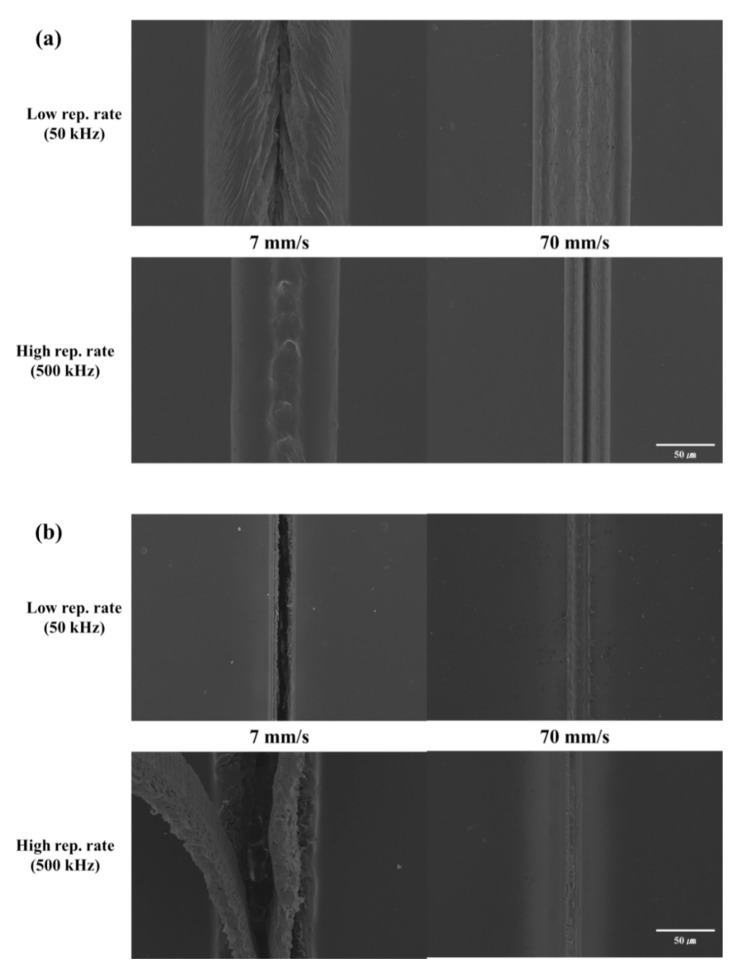
SEM images of laser processing results under stationary beam irradiation with the same pulse energy. (**a**) Polyethylene terephthalate (PET) and (**b**) polyimide (PI).

**Figure 8 polymers-12-02018-f008:**
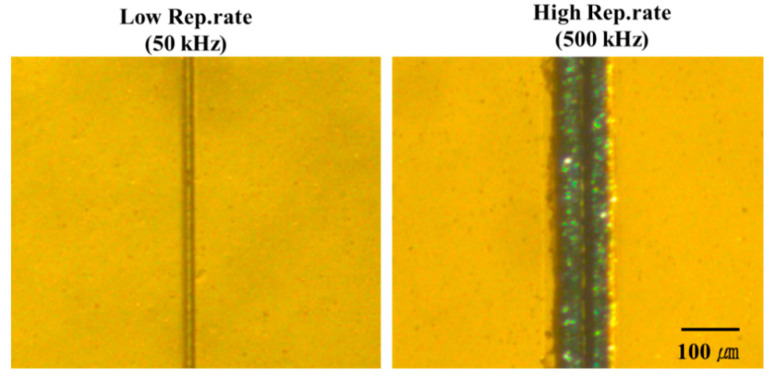
Reflective optical microscopy images of polyimide (PI) samples processed by low and high pulse repetition rates (at 7 mm/s).

**Figure 9 polymers-12-02018-f009:**
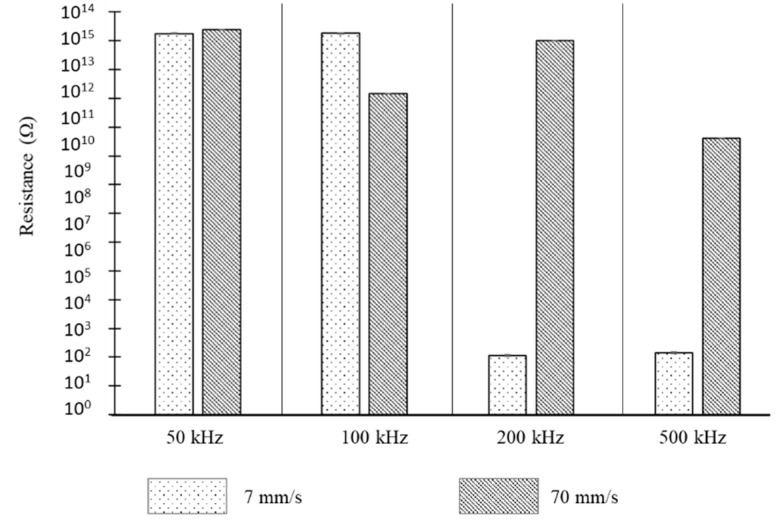
Electrical resistance results and their variation for a PI material with linear absorption using laser scanning processing.

**Figure 10 polymers-12-02018-f010:**
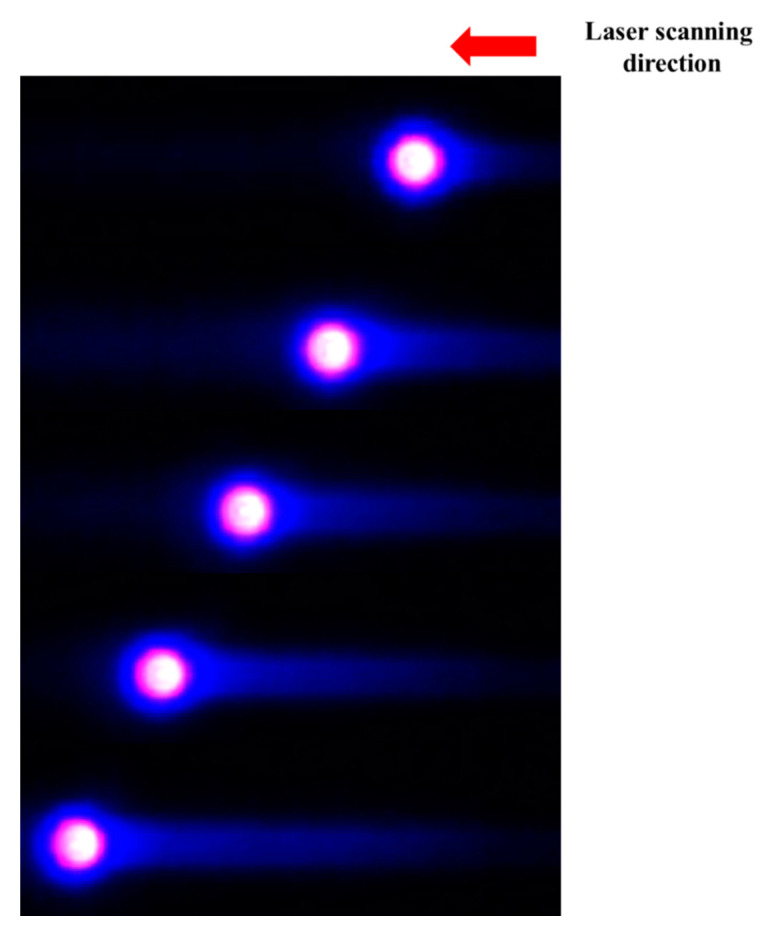
Heat distribution measurement results using an IR camera during laser scan processing.

**Figure 11 polymers-12-02018-f011:**
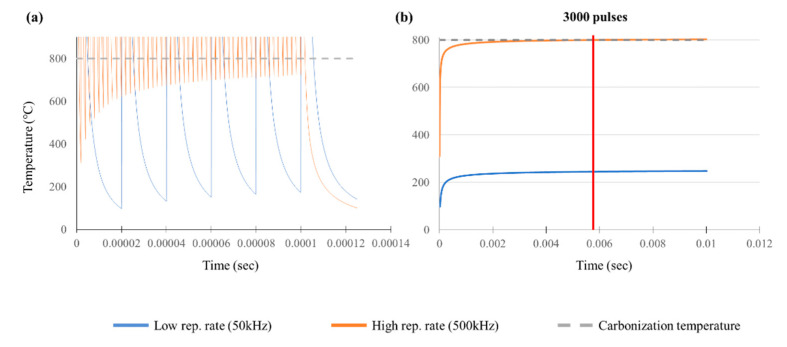
Results of baseline temperature increase due to heat accumulation in PI (**a**) over a few times and (**b**) over more times.

**Table 1 polymers-12-02018-t001:** Comparison of the physical properties of two polymer materials used in the experiment [41,42].

	Polyimide (PI)	Polyethylene Terephthalate (PET)
Density (Kg/m^3^)	1420	1450
Specific heat capacity (J/K·gK)	1090	2030
Thermal conductivity (W/m·K)	0.12	0.29
Thermal diffusivity (m^2^/s)	7.75 × 10^−8^	8.7 × 10^−8^
Melting temperature (°C)	-	200
Carbonization temperature (°C)	800	-

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
