# Peer review of "Surface Characteristics of Polymers with Different Absorbance after UV Picosecond Pulsed Laser Processing Using Various Repetition Rates"

_polymers, 2020, doi:10.3390/polym12092018_

Round 1
Reviewer 1 Report
Review on “Surface Characteristics of polymers with difference absorbance after UV picosecond laser processing using various repetition rates”
Though there exist a plethora of publications that deal with the interaction of pulsed lasers with polymers and its effect on the surface properties, most of these studies are either focused on nanosecond pulsed laser-based ablation or femtosecond pulse-based laser ablation. As there is only a limited number of studies that deal with picosecond laser ablation on polymers, particularly using UV lasers, the article is interesting to accept. However, there require some major changes in the style and format before its acceptance.
- The overall length of the article is high. It can be reduced.
- The introduction contains 47 references. However, very little work is mentioned about the picosecond laser interaction with materials. The number of references can be reduced
- The authors must emphasize the importance of the role of different time scale laser pulses and its interaction with matter in the context of various phenomena happening in the matter (e.g. electron-phonon interaction, electron-electron interaction, etc). Without that the concept of cold ablation and hot ablation have limited meaning
- In the experimental setup (figure 2 b. and c), the authors employed an adhesive. What is the role of the thermal properties of this adhesive in the results explained here?
- Many of the figures are redundant. It can be moved to supplementary information. The total number of figures in the main text can be reduced to 5-8
- There are a few typo errors which have to be rectified in the revised version
- Even in the discussion part, there is a lot of unnecessary explanation that may not suit a journal paper, e.g. difference between linear and nonlinear absorption and subdivision of linear absorption. An appropriate reference can avoid it.
- What is the correlation between the energy employed here and laser ablation threshold of these polymer materials
- In the discussion part, the authors state that “Based on the cited results, it was predicted…” Add appropriate references while making such statements
- The discussion part includes too many unnecessary jargons which can be excluded to make it more clear and crisp.
Author Response
Thank you very much for all the comments from the reviewers. We put our best efforts to address all the issues raised by each reviewer. We also made the appropriate changes in the manuscript where they are needed.
Point 1: The overall length of the article is high. It can be reduced.
Response 1: I agree with some of your points. However, we tried to make it as concise and clear as possible to explain the result
s of picosecond laser processing with samples of different physical and optical properties of the experimented material. As a result, the contents of the text got longer. So, I tried to reduce the contents of the text as much as possible by deleting or modifying the repeated parts of the overall contents. (34-37 lines, 396-399 lines, 405-407 lines, 405-433 lines, 535-541 lines, and 645-654 lines deleted)
Point 2: The introduction contains 47 references. However, very little work is mentioned about the picosecond laser interaction with materials. The number of references can be reduced
Response 2: In agreement with your opinion, we reduced the number and content of reference literature in the introduction from 47 to 37 and applied it to the introduction.
Point 3: The authors must emphasize the importance of the role of different time scale laser pulses and its interaction with matter in the context of various phenomena happening in the matter (e.g. electron-phonon interaction, electron-electron interaction, etc). Without that the concept of cold ablation and hot ablation have limited meaning.
Response 3: I agree with you. The interaction of laser pulses with materials has many phenomena. To that point, we refer to the interaction between the material and the laser in terms of time within the paper. When talking about the perspective of cold ablation in the ultra-short pulsed laser processing, the explanation is given only by the time the energy of the electrons absorbed from the laser is transferred to the lattice.
Point 4: In the experimental setup (figure 2 b. and c), the authors employed an adhesive. What is the role of the thermal properties of this adhesive in the results explained here?
Response 4: In our experiment, we added adhesive in the middle of the sample because the flexibility of the sample makes it difficult to maintain the focus position accurately, so we used it to fix the sample. since the thickness of Polyethylene terephthalate (PET) and Polyimide (PI) is significantly more than the adhesive thickness, the effect of the thermal properties of the adhesive were judged to be negligible.
Point 5: Many of the figures are redundant. It can be moved to supplementary information. The total number of figures in the main text can be reduced to 5-8
Response 5: There are about 6 figures in the results section. The remaining figures in the other sections are closely tied to the points made in those sections. Having them to supplementary section risks eroding the overall cohesiveness of those sections.
Point 6: There are a few typo errors which have to be rectified in the revised version
Response 6: I checked the grammar and syntax in the text and completed the modification.
Point 7: Even in the discussion part, there is a lot of unnecessary explanation that may not suit a journal paper, e.g. difference between linear and nonlinear absorption and subdivision of linear absorption. An appropriate reference can avoid it.
Response 7: We added the reference below in lieu of unnecessary explanation.
- Kannatey-Asibu Jr, E. Principles of laser materials processing; John Wiley & Sons: 2009; Vol. 4.
Point 8: What is the correlation between the energy employed here and laser ablation threshold of these polymer materials.
Response 8: The laser energy used in the experiment is more than twice the ablation threshold of the polymer material.
Point 9: In the discussion part, the authors state that “Based on the cited results, it was predicted…” Add appropriate references while making such statements
Response 9: I have finished marking reference.
Point 10: The discussion part includes too many unnecessary jargons which can be excluded to make it more clear and crisp.
Response 10: I tried to reduce the contents of the text as much as possible by deleting or modifying the repeated parts of the overall contents. (396-399 lines, 405-407 lines, 405-433 lines, 535-541 lines, and 645-654 lines deleted)

Reviewer 2 Report
The paper on UV Picosecond pulsed laser processing of the polymers and its related surface morphology analysis is well written, and dealt with the effect of repetition rate on its machining performance. There are few revisions, if performed, will make this manuscript more worthy according to my knowledge. They are highlighted below.
- In Abstract, what dose “PI” represent? It should be given full spell when it first appears.
- 2. In Introduction, too many references be included in one sentence, and I am not sure they are all related to your study, so that please re-written your introduction by emphasizing the importance of each reference in your paper.
- On Page 6, the detailed view of Figure 3(b) (right figure) seems to be similar as the left one, as such, I do not think it makes any sense for this figure.
- In Section 4 Discussion, it is too long to demonstrate the phenomena that you have found from your experiment (about 9 pages), and some explanations are repeated again and again. So that I think you should re-written this part to make it as clear and meaningful as possible.
- For Figure 10, is there any more information to distinguish the different features in one figure?
- In general, this paper is too long and the authors need to re-written to conclude the key findings from experiment that show the significance of this paper related to the UV Picosecond pulsed laser processing of the polymers.
Author Response
Thank you very much for all the comments from the reviewers. We put our best efforts to address all the issues raised by each reviewer. We also made the appropriate changes in the manuscript where they are needed.
Point 1: In Abstract, what dose “PI” represent? It should be given full spell when it first appears.
Response 1: We added the full spelling of "PI" to the abstract.
Point 2: In Introduction, too many references be included in one sentence, and I am not sure they are all related to your study, so that please re-written your introduction by emphasizing the importance of each reference in your paper.
Response 2: I rewrote the introduction as much as possible with the contents related to this study.
Point 3: On Page 6, the detailed view of Figure 3(b) (right figure) seems to be similar as the left one, as such, I do not think it makes any sense for this figure.
Response 3: We removed schematic from figure 3 and have displayed only the photo image.
Point 4: In Section 4 Discussion, it is too long to demonstrate the phenomena that you have found from your experiment (about 9 pages), and some explanations are repeated and again. So that I think you should re-written this part to make it as clear and meaningful as possible.
Response 4: I agree with some of your points. However, we tried to make it as concise and clear as possible to explain the results of picosecond laser processing with samples of different physical and optical properties of the experimented material. As a result, the contents of the text got longer. So, I tried to reduce the contents of the text as much as possible by deleting or modifying the repeated parts of the overall contents. (396-399 lines, 405-407 lines, 405-433 lines, 535-541 lines, and 645-654 lines deleted)
Point 5: For Figure 10, is there any more information to distinguish the different features in one figure?
Response 5: Figure 10 shows a measurement of the distribution of heat during laser scanning and lists the images observed sequentially during the beam from right to left. So the figures measured over the same time shows that preheating and residual heat are generated, and the overall figures shows that during laser scanning, these preheating and residual heat continue to appear in the direction in which the laser beam is scanned. And I marked the direction of the laser beam in the figure.
Point 6: In general, this paper is too long and the authors need to re-written to conclude the key findings from experiment that show the significance of this paper related to the UV Picosecond pulsed laser processing of the polymers.
Response 6: I tried to reduce the contents of the text as much as possible by deleting or modifying the repeated parts of the overall contents. (34-37 lines, 396-399 lines, 405-407 lines, 405-433 lines, 535-541 lines, and 645-654 lines deleted)

Reviewer 3 Report
Title: “Surface characteristics of polymers with different absorbance after UV Picosecond pulsed laser processing using various repetition rates”
The paper deals with the laser processing of PET and PI by means of ultra-short laser pulses at 355 nm and 10 ps pulse duration.
The work is interesting, and the manuscript is well structured. I have found some minor grammatical mistakes. I would thank the authors to revise the content according to the following comments:
1.- In abstract, define PI.
2.- In Introduction section, please define each acronym before it is used. For instance, PCB, F-PCB.
3.- In introduction section, the statement “Polymer processing using lasers has been researched extensively, and the source mainly used in the industrial field is the CO2 laser” is inaccurate. Authors can find in the scientific literature many works of laser processing of polymers by short and ultra-short laser radiation. For instance, see M. Soldera et al. “Toward High-Throughput Texturing of Polymer Foils for Enhanced Light Trapping in Flexible Perovskite Solar Cells Using Roll-to-Roll Hot Embossing”, Adv. Eng. Mat. 22 (2020) , or A. Stellmacher et al “Fast and cost effective fabrication of microlens arrays for enhancing light out-coupling of organic light-emitting diodes”, Material letters 252 (2020) 268.
4.- In Introduction section, instead of talking about “hot” or “cold ablation” it would be more accurate to be referred as ablation in thermal and non-thermal regime.
5.- In Introduction section, the statement “PI, which undergoes linear absorption, and PET, which does not, using an ultrashort‐pulsed UV wavelength laser” is inaccurate. Authors should state at which wavelength range the linear absorption occurs and why.
6.- Figures 1-(b) and 1-(c) do not provide useful information. This information should be included as text in the draft.
7.- For the calculation of eq. 1, please include the diameter of the laser beam before the microscope objective.
8.- In lines 112-121 it is not clear whether the laser processing with pulse burst from 10 to 15,000 and the scan speed processing was carried out at 50 kHz, 500 kHz or both repetition rates.
9.- Experimental conditions explained between lines 125 and 140 should be referred to the critical frequency. The repetition rate (frequency) at which the cross over from non-thermal to thermal regime is produced is called critical frequency and can be estimated as fcr=Dth/(dlaser)2, where Dth is the thermal diffusivity and dlaser the laser beam diameter at the focal plane, see A. Benayas et al. “Ultrafast laser writing of optical waveguides in ceramic Yb:YAG: a study of thermal and non-thermal regimes”, Applied Physics A 104 (2011) 301-309.
Therefore, thermal diffusivity and diameter of the laser beam stablishes the threshold for thermal and non-thermal accumulation.
I would thank authors to revise experimental conditions, to calculate critical frequency for both materials and to refer these experimental conditions to critical frequency.
10.- In Section 2, authors should include manufacturer and model of the whole equipment used in this work. Also, manufacturer for polymer samples should be included.
11.- In Section 2, authors talk about physical properties of the polymer samples. A table with these properties should be added in the manuscript. Please, include melting temperature, thermal diffusivity, roughness, and optical transmittance at the laser wavelength used, 355 nm.
12.- To assess carbonization or graphitization in the polymer sample, micro-Raman spectroscopy or XPS techniques should be used.
13.- Figure caption for Figure 4 should include laser pulse energy used.
14.- Based on SEM images provided in Figure 4, it is not clear that 50 kHz is in non-thermal regime since all processed samples presented a circular area around the irradiated zone, at least in PET.
15.- For high-repetition-rates, such as 500 kHz, non-linear absorption processes may occur, and the laser processing can be driven by non-linear absorption instead of linear absorption as authors state. Please, assess the non-linear absorption process and justify why linear absorption is happening during the process instead non-linear absorption.
16.- Authors should indicate which the standard deviation of measurements made for electrical resistance and how many measurements they made.
Author Response
Thank you very much for all the comments from the reviewers. We put our best efforts to address all the issues raised by each reviewer. We also made the appropriate changes in the manuscript where they are needed.
Point 1: In abstract, define PI.
Response 1: We added the full spelling of "PI" to the abstract.
Point 2: In Introduction section, please define each acronym before it is used. For instance, PCB, F-PCB.
Response 2: We added the full spelling of "PCB" and “F-PCB” to the abstract.
Point 3: In introduction section, the statement “Polymer processing using lasers has been researched extensively, and the source mainly used in the industrial field is the CO2 laser” is inaccurate. Authors can find in the scientific literature many works of laser processing of polymers by short and ultra-short laser radiation. For instance, see M. Soldera et al. “Toward High-Throughput Texturing of Polymer Foils for Enhanced Light Trapping in Flexible Perovskite Solar Cells Using Roll-to-Roll Hot Embossing”, Adv. Eng. Mat. 22 (2020) , or A. Stellmacher et al “Fast and cost effective fabrication of microlens arrays for enhancing light out-coupling of organic light-emitting diodes”, Material letters 252 (2020) 268.
Response 3: I revised the phrase because I agreed with what you pointed out. Mainly used have been modified to be first used instead.
Point 4: In Introduction section, instead of talking about “hot” or “cold ablation” it would be more accurate to be referred as ablation in thermal and non-thermal regime.
Response 4: We agreed with what the reviewer pointed out and revised it as follows. Modified to thermal (“hot”) ablation and non-thermal (“cold”) ablation.
Point 5: In Introduction section, the statement “PI, which undergoes linear absorption, and PET, which does not, using an ultrashort‐pulsed UV wavelength laser” is inaccurate. Authors should state at which wavelength range the linear absorption occurs and why.
Response 5: I revised it because I agreed that there was not enough explanation in the introduction part. At 355 nm, the laser wavelength we used, PI is linear absorption, but PET is not linear absorption.
Point 6: Figures 1-(b) and 1-(c) do not provide useful information. This information should be included as text in the draft.
Response 6: An explanation has been included in the text (lines 128–152). In addition, figures 1(b) and (c) have been removed per this point.
Point 7: For the calculation of eq. 1, please include the diameter of the laser beam before the microscope objective.
Response 7: Before the objective lens, the beam is 10 mm in diameter. The diameter of these beams is added to the experiment section.
Point 8: In lines 112-121 it is not clear whether the laser processing with pulse burst from 10 to 15,000 and the scan speed processing was carried out at 50 kHz, 500 kHz or both repetition rates.
Response 8: The pulse repetition rate used in the experiment was performed by changing from 50 kHz to 500 kHz. I added the above contents to the text.
Point 9: Experimental conditions explained between lines 125 and 140 should be referred to the critical frequency. The repetition rate (frequency) at which the cross over from non-thermal to thermal regime is produced is called critical frequency and can be estimated as fcr=Dth/(dlaser)2, where Dth is the thermal diffusivity and dlaser the laser beam diameter at the focal plane, see A. Benayas et al. “Ultrafast laser writing of optical waveguides in ceramic Yb:YAG: a study of thermal and non-thermal regimes”, Applied Physics A 104 (2011) 301-309.
Therefore, thermal diffusivity and diameter of the laser beam stablishes the threshold for thermal and non-thermal accumulation.
I would thank authors to revise experimental conditions, to calculate critical frequency for both materials and to refer these experimental conditions to critical frequency.
Response 9: The critical frequency calculated from the reference you recommended was 54 MHz (PI) and 60 MHz (PET), respectively. In the reference literature, the criteria for dividing thermal and nonthermal were divided by normalized frequencies through critical frequency.
Reference is made to nonthermal range machining if the normalized Frequency value is less than 1, and thermal range machining if it is greater than 1. These contents are added to our experiment section.
Point 10: In Section 2, authors should include manufacturer and model of the whole equipment used in this work. Also, manufacturer for polymer samples should be included.
Response 10: I added the manufacturer and model of all the equipment used in the study.
Point 11: In Section 2, authors talk about physical properties of the polymer samples. A table with these properties should be added in the manuscript. Please, include melting temperature, thermal diffusivity, roughness, and optical transmittance at the laser wavelength used, 355 nm.
Response 11: The absorption, which is already an optical characteristic, is shown in figure 2, and is described in the text. And we added physical characteristics in a table.
Point 12: To assess carbonization or graphitization in the polymer sample, micro-Raman spectroscopy or XPS techniques should be used.
Response 12: As you pointed out, we agree to distinguish the carbonization of polymer through micro-Raman spectroscopy or XPS technology. However, if you look at our results, the morphology characteristics of the sample that are assumed to have been carbonization have been identified in the reference literature. And we measured the change in electrical resistance only from samples that are assumed to have been carbonization through the measurement of electrical resistance. In addition, the results of the analysis through the heat conduction equation confirmed that residual heat exists above the carbonization temperature only in conditions where the change in morphology occurs and the change in electrical resistance is observed. I think this observation provides sufficient evidence to say that the Polyimide (PI) has been carbonization.
Point 13: Figure caption for Figure 4 should include laser pulse energy used.
Response 13: 0.8 uJ, the pulse energy used in the experiment, was added to the caption.
Point 14: Based on SEM images provided in Figure 4, it is not clear that 50 kHz is in non-thermal regime since all processed samples presented a circular area around the irradiated zone, at least in PET.
Response 14: In the case of PET, it is written in the text that thermal processing occurs even at 50 kHz. for non-thermal machining it states that only Polyimide (PI), with linear absorption at 355 nm of laser wavelength which we use, occurs at 50 kHz.
Point 15: For high-repetition-rates, such as 500 kHz, non-linear absorption processes may occur, and the laser processing can be driven by non-linear absorption instead of linear absorption as authors state. Please, assess the non-linear absorption process and justify why linear absorption is happening during the process instead non-linear absorption.
Response 15: The text does not mention that non-linear absorption occurs under high pulse repetition rates. It has been modified to specify that linear absorption (in the case of Polyimide (PI)) and nonlinear absorption (in the case of Polyethylene terephthalate (PET)) occurs depending on the polymer.
Point 16: Authors should indicate which the standard deviation of measurements made for electrical resistance and how many measurements they made.
Response 16: Each condition was measured 100 times. The value of the standard deviation per condition has been added to the body. (Written on lines 381 and 382.)

Round 2
Reviewer 1 Report
The authors are accounted the suggestions made on the previous version. The manuscript is now can be accepted in the journal
Author Response
Thank you for your review.
Reviewer 3 Report
Authors have revised the manuscript according to the comments and suggestions. Paper is suitable for being published after the revision of the following minor comments:
- In the revised manuscript authors state that "in the case of PET, the critical frequency is 60 MHz, [...] and in the case of PI, the critical frequency is 54 MHz ...". Please, revise the calculations. Accounting with the provided thermal difusivity and laser diameter these values are in kHz (60 kHz and 54 kHz).
- Please, provide the references for the values provided in Table 1.
Author Response
Thank you very much for all the comments from the reviewers. We put our best efforts to address all the issues raised by each reviewer. We also made the appropriate changes in the manuscript where they are needed.
Point 1: In the revised manuscript authors state that "in the case of PET, the critical frequency is 60 MHz, [...] and in the case of PI, the critical frequency is 54 MHz ...". Please, revise the calculations. Accounting with the provided thermal difusivity and laser diameter these values are in kHz (60 kHz and 54 kHz).
Response 1: As you pointed out, I calculated again and found that the unit was wrong. The value of the recalculated result has been modified. (refer to 160 – 162 lines)
Point 2: Please, provide the references for the values provided in Table 1.
Response 2: We added reference to table 1.
